# Estimation of the Gini coefficient based on two quantiles

**Pingsheng Dai**[1]*, **Sitong Shen**[2]

**1** School of Finance and Economics, Jimei University, Xiamen, Fujian, China, **2** School of Mathematics and Statistics, Nanjing University of Science and Technology, Nanjing, Jiangsu, China

* daips@jmu.edu.cn

## Abstract

Based on the Palma proposition and the Lorenz fitting curve, this paper estimates the sample Gini coefficient using the income share of the top 10% and bottom 40% of the population. Empirical research shows that the absolute error between the estimated value and sample Gini coefficient is within a hundredth. Monte Carlo simulation shows that the new method has good performance and robustness for estimating Gini coefficients with different sample sizes and different inequality levels. Using the two quantiles in the deciles to estimate the sample Gini coefficient and the Lorenz fitting curve is a practical method.

**Data Availability Statement:** The datasets generated and/or analyzed during the current study inclued in this article can be accessed from the WIID datasets of the United Nations University. (https://www.wider.unu.edu/database/world-income-inequality-database-wiid).

## 1 Introduction

In recent years, income-share-based limited data has become an important way for some economies and international institutions to store and publish data. Policymakers are more likely to want to understand the Gini coefficient that reflects income inequality when formulating public policies. There are two main methods for estimating the income Gini coefficient using limited data. One is to fit the income distribution function and then calculate the Gini coefficient. Some scholars have proposed fitting methods for several types of income distributions, such as lognormal distribution, gamma distribution, Beta-2 distribution, and generalized Pareto distribution [1–4]. The second is to fit the income Lorenz curve. The fitting curve is composed of points with cumulative population share and cumulative income share as horizontal and vertical coordinates. The quantile data just constitute the points on the Lorenz fitting curve, including the two endpoints of the $45^0$ line. The quantile approach provides great convenience for fitting the Lorenz curve.

The existing Lorenz curves include single-parameter [5–8], two-parameter [9–11], three-parameter [12, 13], and four-parameter [14, 15] forms. Since only some functions of the above curves satisfy the non-negativity requirements of the first and second-order derivatives [9, 13, 16], we believe that it is more appropriate to call these curves **Lorenz** fitting curves.

Chotikapanich and Griffiths [17] proposed a likelihood function method for parameter estimation of the Lorenz curve. This method assumes that the income shares of each quantile follow a joint Dirichlet distribution, adding an adjustment parameter λ related to the variance to the Lorenz fitting curve to obtain parameter estimates through the maximum likelihood

**Funding:** Dai P's study was funded by a grant (#20BJY238) from the National Social Science Fund of China (http://www.nopass.gov.cn/).

**Competing interests:** The authors have declared that no competing interests exist.

(ML) method:

$$
\begin{aligned}
\log[f(\boldsymbol{q}|\boldsymbol{\theta})] \quad &= \log \Gamma(\lambda) + \sum_{i=1}^{N} \{\lambda[L(p_i, \boldsymbol{\theta}) - L(p_{i-1}, \boldsymbol{\theta})] - 1\} \times \log q_i \\
&\quad - \sum_{i=1}^{N} \log \Gamma\{\lambda[L(p_i, \boldsymbol{\theta}) - L(p_{i-1}, \boldsymbol{\theta})]\}
\end{aligned}
\tag{1}
$$

where $f$ is the Dirichlet joint distribution function, $\lambda$ is the unknown auxiliary parameter, $\boldsymbol{\theta}$ is a parameter vector, $p_i$ is the cumulative population share to the $i$-th quantile, $q_i$ is the income share of the $i$-th decile ($i = 1, \ldots, N$), $\boldsymbol{q} = (q_1, q_2, \ldots, q_N)^T$, $N$ is the number of quantiles, $\Gamma$ and $L$ represent the gamma function and the Lorenz curve, respectively.

Some scholars have found that although Chotikapanich and Griffiths [17]'s method has a good fitting effect [8], increasing the number of parameters will complicate the estimation. In fact, the parameter estimate can be obtained directly by minimizing the sum of squares of the errors between the cumulative income share and the fitted one (referred to as the error minimization technique) [18]. It is also difficult to verify whether the income shares of each quantile follow the joint Dirichlet distribution, which may lead to large deviations in parameter estimates and affect the estimation of the Gini coefficient. The error minimization technique can be expressed as follows:

$$
\hat{\theta} = \min_{\theta} \sum_{i=1}^{N} [y_i - L(x_i, \theta)]^2
\tag{2}
$$

where $\boldsymbol{\theta}$ is a parameter vector, $y_i$ and $x_i$ represent the cumulative income and cumulative population shares of the $i$-th quantile of data ($i = 1, \ldots, N$), respectively.

Given that the maximum likelihood method of Chotikapanich and Griffiths [17] could not be applied to the case of negative income share in the quantile data, Lee and Suh [19] argue that existing methods of removing negative values or replacing them with zero underestimate inequality levels, Shen and Dai [16] proposed a linear regression method based on the Kakwani's three-parameter Lorenz curve [12], the method is better than the error minimization technique of model (2) for estimating the Gini coefficient of economies with medium and high inequality.

It is usually difficult to estimate the parameters of the Lorenz curve using the model (1) or model (2) based on two quantile points to obtain an effective estimate of the sample Gini coefficient. In Palma [20, 21], the author believes that changes in income inequality depend on changes in the income shares of the top 10% and bottom 40% of the population. Cobham and Sumner [22] supported Palma's proposition through regression analysis based on the decile of the World Bank's PovcalNet data. It has become an interesting and challenging problem that how to construct a reasonable Lorenz fitting curve and estimate the sample Gini coefficient based on the income share of the top 10% and bottom 40% of the population.

The contributions of this paper are mainly three-fold: firstly, based on the Lorenz curve of Kakwani [12], a simplified two-parameter Lorenz fitting curve is proposed, and the income share of the bottom 40% and the top 10% of the population is used to obtain an effective estimate of the sample Gini coefficient. The Lorenz fitting curve is estimated in the most parsimonious (least conditional) way. Secondly, the best pairwise method bridges the Gini coefficient and the Palma ratio, because both of them are determined by the same income shares and have equivalence. Thirdly, the best pairwise method provides governments and policymakers with a convenient and low-cost tool to access information on inequalities. This is because policymakers only need to obtain the income shares of the bottom 40% and the top 10% of the

population to obtain an accurate measure of the Gini coefficient, without having to obtain full information on the distribution of income.

The remaining of this paper is organized as follows: Section 2 introduces the parameter estimation of the Lorenz fitting curve and proposes the best pairwise proposition, Section 3 tests the new proposition through Monte Carlo simulations and diversified income distribution data, Section 4 summarizes the conclusions.

## 2 Materials and methods

### 2.1 Ethical approval

Ethical approval was not required as the study did not involve human participants.

### 2.2 Two-parameter Lorenz fitting curve

Many studies have found that the Lorenz curve provided by Kakwani [12] generally performs better than other forms of Lorenz curves in empirical studies [11, 16, 23, 24]. Therefore, this paper starts with this curve. The form of the Lorenz curve given by Kakwani [12] is as follows:

$$L(x, a, p, q) = x - ax^p(1 - x)^q \qquad a > 0, 0 \le p \le 1, 0 \le q \le 1 \tag{3}$$

where, $x$ represents the cumulative population share, $a$ is a scale parameter, $p$ and $q$ are shape parameters, $L$ denotes the cumulative income share. Sarabia et al. [13] criticized that model (3) does not meet the theoretical conditions of the Lorenz curve. For example, if the parameters $a$=1, $p$=0.96, $q$=0.68, and $x$=0.1, the value of the Lorenz curve of Eq (3) is -0.0021, that is, the value of the function is less than 0. With this in mind, it may be more appropriate to call model (3) a Lorenz fitting curve rather than a Lorenz curve.

Schader and Schmid [15] extended the model (3) to a four-parameter model in Eq (4).

$$L(x, p, q) = x^r - x^p(1 - x)^q \tag{4}$$

where, Schader and Schmid [15] deliberately ignore the theoretical boundaries of parameter values. They find that the model (4) has a better performance than the model (3), since the sum of squares of the deviations between true and fitting values is smaller under model (4). In fact, determining the parameter value boundary according to the theoretical conditions of the Lorenz curve has brought great trouble to empirical studies. For instance, Sitthiyot and Holasut [25] find that conditional on the parameter value boundary of the single-parameter Lorenz curve proposed by Paul and Shankar [8], in many cases, an effective estimation of the Gini coefficient cannot be obtained. Whether it is to estimate the points on the Lorenz curve or to estimate the corresponding Gini coefficient, models (3) and (4) show that the Lorenz fitting curve may not be a Lorenz curve. The fitting curve only needs to pass through points (0, 0) and (1, 1), and there is no need to determine the parameter value boundary of the fitting curve in advance. We find that this method can eliminate the boundary trouble of parameter values encountered by Sitthiyot and Holasut [25].

Considering that the three parameters cannot be determined with two quantiles, the following changes have been made to model (3):

$$L(x, p, q) = x - x^p(1 - x)^q \qquad p > 0, q > 0 \tag{5}$$

Model (4) can be transformed into the following form:

$$\ln[x - L(x, p, q)] = p\ln x + q\ln(1 - x) \tag{6}$$

Therefore, the Lorenz fitting curve is determined by two quantiles, which is equivalent to solving the following system of simultaneous equations:

$$\begin{cases} \ln(x_1 - y_1) & = p\ln(x_1) + q\ln(1 - x_1) \\ \ln(x_2 - y_2) & = p\ln(x_2) + q\ln(1 - x_2) \end{cases} \tag{7}$$

where, $(x_1, y_1)$ and $(x_2, y_2)$ represent the coordinates of two decile points on the Lorenz fitting curve. The solutions to the two parameters of equation system (6) are:

$$\begin{aligned} p &= \frac{\ln(x_1 - y_1)\ln(1 - x_2) - \ln(x_2 - y_2)\ln(1 - x_1)}{\ln x_1 \ln(1 - x_2) - \ln x_2 \ln(1 - x_1)} \\ q &= \frac{\ln x_1 \ln(x_2 - y_2) - \ln x_2 \ln(x_1 - y_1)}{\ln x_1 \ln(1 - x_2) - \ln x_2 \ln(1 - x_1)} \end{aligned} \tag{8}$$

The formula of the Gini coefficient corresponding to parameters $p$ and $q$ is:

$$G = 2\left(\frac{1}{2} - \int_0^1 [x - x^p(1-x)^q]dx\right) = 2Beta(p + 1, q + 1) \tag{9}$$

## 2.3 Validity criteria for estimating the Gini coefficient

The purpose of the new method is to estimate the deciles of data and the sample Gini coefficient using only two quantiles. First, to judge the validity of the estimated sample Gini coefficient, this paper uses the absolute error between the estimated value and the true Gini coefficient as a measure for the judgement. Specifically, when the absolute error is within a hundredth, the estimation is considered to be valid. The root mean square error (*RMSE*) is used for the comparison of multiple estimation errors. The smaller the *RMSE*, the better the estimation. Secondly, for fitting the deciles of data, the information inaccuracy measure (*IIM*) index is used. For fitting the Lorenz curve, the mean square error (*MSE*) index is used. The calculation formulas for the two indices are as follows:

$$IIM = \sum_{i=1}^N q_i \ln\left(\frac{q_i}{\hat{q}_i}\right), \quad MSE = \frac{1}{N}\sum_{i=1}^N \left[y_i - L(x_i, \hat{\theta})\right]^2 \tag{10}$$

where, $N$ is the number of quantiles, $q_i$ is the income share of the $i$-th quantile, $\hat{q}_i$ is the estimated income share ($i = 1, 2, \ldots N$) [8, 11]. The smaller the absolute value of the fitting index (i.e., *IIM* or *MSE*), the better the fitting.

## 2.4 Best pairwise proposition

The deciles of data are most common ($N$=10), so consider available pairings among 10%, 20%, 30%,..., 70%, 80%, and 90% quantiles, there are 36 combinations in total. For example, we can take a data on each side of the fifth decile point, which corresponds to the 4th and 9th deciles on the Lorenz curve. We can search the best pairs among possible matches to estimate the Gini coefficient. Dagum [26] believes that a good parametric form of the Lorenz curve must be able to characterize the income distribution in different regions, different socio-economic forms and different periods. This paper divides the income distribution situation into four categories according to the level of the sample Gini coefficient: low income inequality ($0 \le$ Gini coefficient $< 0.3$), medium income inequality ($0.3 \le$ Gini coefficient $< 0.4$), high income

inequality ($0.4 \leq$ Gini coefficient $< 0.5$), and very high income inequality ($0.5 \leq$ Gini coefficient $< 1$).

Records of the World Income Inequality Database (WIID) are divided into four categories based on data quality: high, average, low and not known levels. To ensure data quality, we extract the decile income share data of 16 economies that are with different socioeconomic backgrounds and inequality levels from the World Bank's PIP records in the WIID database (The quality level of PIP is average). Then we estimate the Gini coefficient using the above-mentioned pairing method, and compare the estimated Gini coefficients with the sample Gini coefficients reported in the database to find the best pairing and propose new proposition. The calculation results are reported in Table 1.

According to the validity criteria for estimating the Gini coefficient, Table 1 suggests there are two best pairings, i.e. 40% and 90% pair, and 30% and 80% pair. The absolute errors of the Gini coefficient estimates of the two pairs are within a hundredth, regardless of the maximum error or the minimum error. Based on the comparison of the estimated errors of 16 economies, the *RMSE* value of the (40%, 90%) pairing is 0.00196, which is smaller than the *RMSE* value (0.00267) of the (30%, 80%) pairing, indicating that the (40%, 90%) pairing is better than the (30%, 80%) pairing. The Palma ratio is an important measure of income inequality (Cobham and Sumner, 2002; Palma, 2014), which is equal to the ratio of income shares of the top

**Table 1. Estimated errors of the Gini coefficient for the main pairwises of decile of 16 economies.**

| Year | Economies | True Gini | Errors between observed and estimated Gini under various pairings | | | | | |
|------|-----------|-----------|------------|------------|------------|------------|------------|------------|
| | | | (30%,80%) | (40%,90%) | (50%,90%) | (60%,40%) | (70%,30%) | (80%,40%) |
| 2020 | Belgium | 0.2540 | -0.00062 | 0.00364 | 0.02208 | 0.02695 | 0.01122 | 0.01381 |
| 2019 | Czechia | 0.2402 | -0.00132 | 0.00027 | 0.01943 | 0.03002 | 0.01314 | 0.01251 |
| 2018 | Slovakia | 0.2097 | -0.00017 | -0.00197 | 0.02008 | 0.03546 | 0.01698 | 0.01462 |
| 2017 | Iceland | 0.2487 | -0.00122 | -0.00173 | 0.01981 | 0.03066 | 0.01420 | 0.01225 |
| 2020 | Austria | 0.3802 | 0.00081 | 0.00185 | 0.01990 | 0.03187 | 0.01482 | 0.01390 |
| 2019 | Albania | 0.3430 | -0.00274 | 0.00119 | 0.01411 | 0.01975 | 0.00747 | 0.00721 |
| 2018 | Germany | 0.3159 | -0.00005 | 0.00182 | 0.01367 | 0.02025 | 0.00924 | 0.00855 |
| 2017 | Russia | 0.3670 | -0.00012 | 0.00203 | 0.01366 | 0.01887 | 0.00816 | 0.00794 |
| 2020 | France | 0.4230 | 0.00060 | 0.00142 | 0.01235 | 0.02050 | 0.00990 | 0.00830 |
| 2019 | Colombia | 0.4810 | 0.00152 | 0.00250 | 0.01208 | 0.01857 | 0.00904 | 0.00877 |
| 2018 | USA | 0.4709 | 0.00047 | 0.00162 | 0.00581 | 0.00890 | 0.00418 | 0.00392 |
| 2017 | Malaysia | 0.4107 | 0.00068 | 0.00073 | 0.00914 | 0.01667 | 0.00857 | 0.00665 |
| 2019 | Panama | 0.5150 | 0.00237 | 0.00054 | 0.00469 | 0.01109 | 0.00724 | 0.00480 |
| 2018 | Brazil | 0.5400 | -0.00130 | -0.00026 | 0.00766 | 0.00737 | 0.00622 | 0.00353 |
| 2017 | South Africa | 0.6170 | 0.00245 | 0.00260 | 0.00524 | 0.00690 | 0.00584 | 0.00393 |
| 2016 | Hong Kong | 0.5390 | -0.00925 | 0.00323 | -0.00427 | -0.01963 | -0.01699 | -0.01060 |
| **Error** | Max | | 0.00245 | 0.00364 | 0.02208 | 0.03546 | 0.01698 | 0.01462 |
| | Min | | -0.00925 | -0.00197 | -0.00427 | -0.01963 | -0.01699 | -0.01060 |
| RMSE | All | | 0.00267 | 0.00196 | 0.01408 | 0.02198 | 0.01090 | 0.00956 |
| | Low | | 0.00095 | 0.00225 | 0.02037 | 0.03092 | 0.01404 | 0.01333 |
| | Medium | | 0.00143 | 0.00175 | 0.01556 | 0.02330 | 0.01034 | 0.00976 |
| | High | | 0.00091 | 0.00169 | 0.01020 | 0.01675 | 0.00823 | 0.00716 |
| | Very High | | 0.00497 | 0.00209 | 0.00562 | 0.01235 | 0.01017 | 0.00639 |

[1]The income distribution situation is devided into four categories: low, medium, high and very high.

[2]There are a total of $C_9^2 = 36$ quantile pairs, and only results of the main 6 pairwises are listed in the table.

10% and the bottom 40% groups. The Palma proposition is the theoretical basis for estimating the Gini coefficient using two quantiles. Therefore, this paper extracts the following proposition.

**Definition 1** If the absolute error between the estimated value and the true value of the sample Gini coefficient is less than 0.01, the estimate of the sample Gini coefficient is said to be a valid estimate.

**Proposition 1** The pair of income shares of the bottom 40% and the top 10% groups is the best pairwise for estimating the sample Gini coefficient, since the pair can obtain a valid estimate of the sample Gini coefficient.

Since the income share of the top 10% of the population is equal to 1 minus the income share of the bottom 90% of the population, the above proposition is equivalent to that the pair of the income share of the bottom 40% and the bottom 90% of the population is the best. This method is also called the best pairwise method. From Table 1, it can also be found that the difference between the maximum error and the minimum error of the (40%, 90%) pairing is 0.00561, which is obviously smaller than the 0.01170 of the (30%, 80%) pairing. The smaller range indicates that the former is more robust. As for the RMSE, we find that the RMSE of the Gini coefficient estimated by the (30%, 80%) pairing is relatively small for the low, medium and high inequality groups, and the RMSE of the (40%, 90%) pairing is relatively small for the whole sample and very high inequality groups. Whether that this new method has a general significance can be verified through Monte Carlo simulation. It also can be verified by comparing the estimated Gini coefficients with the reported Gini coefficients in the records of World Bank's PIP and the Luxembourg Income Institute's LIS of the WIID database.

The best pairwise method for estimating the Gini coefficient can be decomposed into the following steps: The first step is to substitute the two income shares (40%, 90%) into the simultaneous Eq (7) to solve the parameters. The second step is to substitute the parameter values into Eq (5) to determine the Lorenz fitting curve. The third step is to calculate the sample Gini coefficient by Eq (9). The calculation of the Gini coefficient requires the help of the Beta function. The Beta function value can be easily realized by commonly used software, such as R, STATA, etc., so the best pairwise method is very practical.

In this paper, with the exception of the error minimization technique, which uses R, the rest of the calculations are performed in Eviews 10.0.

## 3 Results and discussion

### 3.1 Monte Carlo simulation

In order to analyze and verify the sample performance of the best pairwise method, we consider two cases. One is that income follows a log-normal distribution [25]. The other one is that income follows a Beta-2 distribution [3].

**3.1.1 Case of income log-normal distribution.**   First, similar to Arcarons and Calonge [27], we assume that income is generated from the following log-normal distribution:

$$y_t = \exp[9.1171 + 1.1021 \times U(0, 1)] \tag{11}$$

The sample sizes are 500, 1000, 5000, and 10000 respectively. Taking a sample size of 500 as an example, we explain how to generate deciles of data by the following steps:

Step 1 we use formula (11) to generate a random sample of income with a size of 500, and sort sample incomes from small to large to calculate the total income and sample Gini coefficient.

Step 2 starting from the minimum income, for each sample income level, we calculate the number of incomes and the sum of incomes until that sample income (including that sample income), and calculate the cumulative population share and the cumulative income share.

Step 3 we extract the corresponding cumulative income shares for each decile cumulative population share, thus obtaining a set of decile income share.

In order to avoid the possible contingency of a set of data, we repeat the above process 1000 times to obtain 1000 sets of deciles of data.

For each set of deciles of data, a paired test is performed to calculate the estimated error of the Gini coefficient. Then, the minimum error, maximum error and overall *RMSE* are calculated based on the estimated errors of 1000 sets of deciles of data. The other sample sizes are processed similarly, and the results in Table 2 can be obtained. In theory, $\ln y$ obeys the above-mentioned normal distribution, i.e. $\ln y \sim N(9.1171, 1.1021^2)$, and the corresponding theoretical value of the Gini coefficient is $G = 2\Phi(1.1021/\sqrt{2}) - 1 = 0.56420$. However, the error here refers to the deviation between the estimated value and the sample Gini coefficient.

We examine the error between the estimated value and the true value of the sample Gini coefficient under different sample sizes, and look for the best pairs based on the maximum and minimum errors according to the validity criterion. Both the absolute values of the maximum error and minimum error of the (40%, 90%) pair and the (50%, 90%) pair in Table 2 are within a hundredth, so both these pairs are the best matches.

From Table 2, we can find that when the sample size is greater than or equal to 500, the estimates of the Gini coefficient are valid under the pairings (40%, 90%) and (50%, 90%). The RMSE index for the pairing (50%, 90%) is smaller than that of the pairing (40%, 90%) under each kind of sample size, so the estimation performance of the Gini coefficient of the pairing (50%, 90%) is slightly better than that of the pairing (40%, 90%). This shows that different pairings may have their own advantages under different income distributions. As one of the two best pairs, the pair (40%, 90%) provides empirical support for the Palma proposition,

**Table 2. Estimation error of the Gini coefficient for the main pairwises of log-normal random income under different sample sizes.**

| Obj. | Error | Pairwises estimation error (1000 repetitions) | | | | | |
|---|---|---|---|---|---|---|---|
| | | (30%,80%) | (40%,90%) | (50%,90%) | (60%,40%) | (70%,30%) | (80%,40%) |
| **500** | Max | 0.00345 | 0.00998 | 0.00920 | 0.01607 | 0.01003 | 0.00635 |
| | Min | -0.01334 | -0.00729 | -0.00889 | -0.03014 | -0.02077 | -0.01349 |
| | RMSE | 0.00560 | 0.00366 | 0.00276 | 0.00946 | 0.00856 | 0.00518 |
| **1000** | Max | 0.00141 | 0.00851 | 0.00546 | 0.00752 | 0.00184 | 0.00200 |
| | Min | -0.01294 | -0.00371 | -0.00851 | -0.02166 | -0.01663 | -0.01222 |
| | RMSE | 0.00494 | 0.00311 | 0.00207 | 0.00800 | 0.00752 | 0.00447 |
| **5000** | Max | -0.00117 | 0.00585 | 0.00194 | 0.00063 | -0.00163 | -0.00099 |
| | Min | -0.00740 | -0.00044 | -0.00300 | -0.01292 | -0.01197 | -0.00687 |
| | RMSE | 0.00447 | 0.00258 | 0.00096 | 0.00676 | 0.00686 | 0.00394 |
| **10000** | Max | -0.00241 | 0.00407 | 0.00137 | -0.00170 | -0.00335 | -0.00195 |
| | Min | -0.00630 | 0.00088 | -0.00223 | -0.01067 | -0.00987 | -0.00579 |
| | RMSE | 0.00439 | 0.00255 | 0.00069 | 0.00651 | 0.00674 | 0.00385 |

[1]The random incomes are generated from $y_t = \exp[9.1171 + 1.1021 \times U(0, 1)]$.

[2]There are a total of $C_9^2 = 36$ quantile pairs, and only results of the main 6 pairwises are listed in the table.

indicating that it is valid to estimate the Gini coefficient using the income share of the lowest 40% and the highest 90% of the population.

**3.1.2 Case of income Beta-2 distribution.** The income distribution function of Hong Kong in 1993 fitted by Chotikapanich et al. [3] is expressed as:

$$y_t = bT/(1 - T), \quad b = 2958.5740, \quad T = Beta(8.6944, 2.0609) \tag{12}$$

Similar to the treatment of lognormal distribution, we examine the error between the estimated value and the true value of the Gini coefficient under different sample sizes. We find that when the sample size is greater than or equal to 500, the pairs (40%, 90%) is **only** the best pairings for the sample Gini coefficient estimation. Since the sample size of household surveys is usually more than several thousand, the results in Table 3 verify the correctness of Proposition 1, and provide empirical support for the Palma Proposition.

We also try to have a small sample size, for example, when the sample size is equal to 100 and 200 respectively, the estimation accuracy of the sample Gini coefficient decreases, and some results that do not meet the effective estimation will occur.

According to Tables 2 and 3, we find that as the sample size increases, the *RMSE* of the best pairing also shows an overall downward trend (i.e. the estimation accuracy improving), indicating that the sample size also has an important impact on the performance of the best pairwise method. In summary, the income distribution form and sample size can play an important role in the validity of the best pairwise method.

**3.1.3 Performance of the method when estimating deciles of data.** We have found that the best pairwise (40%, 90%) method is valid for estimating the Gini coefficient under decile case. So, conversely, how does the best pairwise method perform when estimating deciles of data? Below we compare the best pairwise method with the Sitthiyot and Holasut [11]'s method (hereinafter referred to as SH method). The SH method estimates the deciles of data based on the Gini coefficient and the income shares of the bottom 10% and top 10% of the population, while the best pairwise method is a method based on the income shares of the bottom 40% and top 10% of the population.

**Table 3. Estimation error of the Gini coefficient for the main pairs of Beta-2 distribution random income under different sample sizes.**

| Obj. | Error | Pairwises estimation error (1000 repetitions) | | | | | |
|------|-------|-----------|-----------|-----------|-----------|-----------|-----------|
| | | (30%,80%) | (40%,90%) | (50%,90%) | (60%,40%) | (70%,30%) | (80%,40%) |
| **500** | Max | 0.01237 | 0.00905 | 0.01844 | 0.04762 | 0.03010 | 0.01998 |
| | Min | -0.00717 | -0.00606 | 0.00565 | 0.01072 | 0.00125 | 0.00204 |
| | RMSE | 0.00281 | 0.00216 | 0.01249 | 0.02658 | 0.01328 | 0.00918 |
| **1000** | Max | 0.01246 | 0.00450 | 0.01724 | 0.04352 | 0.02865 | 0.01905 |
| | Min | -0.00489 | -0.00390 | 0.00823 | 0.01449 | 0.00353 | 0.00312 |
| | RMSE | 0.00219 | 0.00153 | 0.01261 | 0.02701 | 0.01357 | 0.00942 |
| **5000** | Max | 0.01363 | 0.00346 | 0.01573 | 0.04039 | 0.02857 | 0.01903 |
| | Min | -0.00176 | -0.00149 | 0.01098 | 0.02051 | 0.00842 | 0.00667 |
| | RMSE | 0.00158 | 0.00081 | 0.01281 | 0.02731 | 0.01372 | 0.00960 |
| **10000** | Max | 0.00719 | 0.00209 | 0.01486 | 0.03850 | 0.02355 | 0.01562 |
| | Min | -0.00110 | -0.00115 | 0.01143 | 0.02360 | 0.01036 | 0.00762 |
| | RMSE | 0.00125 | 0.00066 | 0.01282 | 0.02736 | 0.01370 | 0.00959 |

[1]The random incomes are generated from $y_t = bT/(1 - T)$, $b = 2958.5740$, $T = Beta(8.6944, 2.0609)$.

[2]There are a total of 36 quantile pairs, and only results of the main 6 pairwises are listed in the table.

The following is a comparative analysis on the goodness of fit of estimating deciles of data based on the two methods. The needed Lorenz curve of the SH method when estimating decile data is as following:

$$L(x, k, p) = (1-k)x^p + k\left[1 - (1-x)^{\frac{1}{p}}\right]$$ (13)

The two parameters $k$ and $p$ can be obtained by the following calculation formula:

$$p = \frac{1+G}{1-G} \geq 1, \quad k = \frac{a - r + c \times r}{c \times r - r + d \times r + a + b - 1}$$ (14)

where, $G$ is the sample Gini coefficient, $a = 0.1^P$, $b = 0.9^P$, $c = 0.9^{1/p}$, $d = 0.1^{1/p}$, and $r = L_{0.1}/(1 - L_{0.9})$.

Below, taking the sample size as 10000, we use models (11) and (12) to generate a decile data (as the true value) and calculate the sample Gini coefficient. The sample Gini coefficient under the log-normal distribution is equal to 0.56438, and it is equal to 0.47397 under the income Beta-2 distribution. Based on the above data, the best pairwise and SH methods are used to estimate the decile data. Table 4 reports the estimated decile data and the goodness of fit indicators MSE and IIM under two income distributions.

In order to judge the validity of the pairwise method, we compares the fit-goodness indicators of the best pairwise and SH methods. For the MSE indicator, the MSE of the best pairwise method is always smaller than that of the SH method under both distribution functions, indicating that the best pairwise method is more advantageous. In terms of the IIM index, the best pairwise method is more advantageous under the lognormal distribution, but less advantageous in the Beta-2 distribution. In general, the best pairwise and the SH method have their own advantages in fitting decile data, indicating that the applicability of the two methods is also closely related to the income distribution. However, the best pairwise method only uses

**Table 4. Comparison of the best pairwise and SH methods in estimating deciles of data under two distributions.**

| Variable | Log-normal distribution | | | Variable | Beta-2 distribution | | |
|---|---|---|---|---|---|---|---|
| | True Decile | Est. of income share | | | True Decile | Est. of income share | |
| | | Pairwise method | SH method | | | Pairwise method | SH method |
| D1 | 0.00865 | 0.00952 | 0.00743 | D1 | 0.02426 | 0.04452 | 0.02218 |
| D2 | 0.01708 | 0.01577 | 0.00997 | D2 | 0.03343 | 0.03199 | 0.02705 |
| D3 | 0.02557 | 0.02513 | 0.01621 | D3 | 0.04049 | 0.03221 | 0.03476 |
| D4 | 0.03549 | 0.03637 | 0.02750 | D4 | 0.04810 | 0.03755 | 0.04518 |
| D5 | 0.04716 | 0.04977 | 0.04504 | D5 | 0.05702 | 0.04703 | 0.05840 |
| D6 | 0.06358 | 0.06611 | 0.06995 | D6 | 0.06812 | 0.06090 | 0.07471 |
| D7 | 0.08476 | 0.08703 | 0.10346 | D7 | 0.08205 | 0.08051 | 0.09485 |
| D8 | 0.11587 | 0.11625 | 0.14725 | D8 | 0.10329 | 0.10940 | 0.12075 |
| D9 | 0.17342 | 0.16563 | 0.20517 | D9 | 0.14627 | 0.15890 | 0.15918 |
| D10 | 0.42842 | 0.42842 | 0.36802 | D10 | 0.39698 | 0.39698 | 0.36296 |
| goodness of fit | | | | goodness of fit | | | |
| MSE | | 0.00002 | 0.00068 | MSE | | 0.00018 | 0.00025 |
| IIM | | 0.00044 | 0.01860 | IIM | | 0.01002 | 0.00597 |

[1]The random incomes of log-normal distribution are generated from $y_t = \exp[9.1171 + 1.1021 \times U(0, 1)]$.

[2]The random incomes of Beta-2 distribution are generated from $y_t = bT/(1-T)$, $b = 2958.5740$, $T = Beta(8.6944, 2.0609)$.

the income shares of the bottom 40% and top 10% of the population, requiring fewer data conditions.

## 3.2 Discussion

We have shown that the (40%, 90%) best pairwise method is effective in estimating the sample Gini coefficient and performs well in fitting deciles of data. In the Monte Carlo simulation, we verify the validity of the best pairwise method under two specific income distributions. However, it is usually impossible to know in advance the types of income distribution and the specific distribution parameters. Next, we will directly verify the effectiveness of the best pairwise method by using two sets of records in the WIID database, one is obtained from the World Bank's PIP records, and the other from the Luxembourg Income Institute's LIS records.

**3.2.1 Applicability of the best pairwise method to PIP and LIS records.** One of the reasons for using the PIP and LIS records is the higher quality of these decile data, the PIP data and LIS data are average and high, respectively. The second is that these two sets of records reported the sample Gini coefficient for different economies, facilitating comparative analysis. In November 2023, there are 2383 and 5545 records of decile data in WIID database, which is collected from PIP and LIS databases. As mentioned earlier, the best pairwise method can estimate the Gini coefficient without requiring the distribution information of data. In reality, it is difficult to know in advance what kind of distribution income and wealth follow, the best pairwise method should have good practicality. We will verify the applicability of the best pairwise method using the above-mentioned decile datasets. Similar to the inequality classifications of economies in Table 1, we divide the sample economies into four categories, i.e. low inequality, medium inequality, high inequality, and very high inequality. We first estimate the values of the Gini coefficient by using the best pairwise and the error minimization methods, then calculate the maximum and minimum errors between the estimated and the true value of Gini coefficients for each category, the root mean square error (RMSE) as well. Table 5 reports the results.

As shown in Table 5, the maximum error, the minimum error, and the root mean square error are all within a hundredth, indicating that the income shares of the bottom 40% and top 10% of the population can be used to obtain an effective and robust estimate of the Gini coefficient.

**3.2.2 Comparison between the best pairwise and the error minimization methods.** It can be found that when using the error minimization technique, the Gini coefficient estimation of models (3) and (4) are both valid, since the absolute values of the maximum and minimum errors are within a hundredth. The error minimization method based on models (3) and (4) are both better than the (40%, 90%) pairwise method in the entire sample. For model (3), although both are effective estimates of the Gini coefficient, the (40%, 90%) pairwise method is better than the error minimization method only in the high inequality category. For model (5), the error minimization technique cannot obtain an effective estimate of the Gini coefficient in multiple inequality categories. Despite using all the information of the decile data, the maximum error of the error minimization technique in multiple inequality categories is greater than 0.01. Therefore, when using model (5), the performance of the best pairwise method is better than the error minimization method under most inequality categories.

## 4 Conclusions

Based on the Palma proposition, this paper proposes an best pairwise method to estimate the Gini coefficient, and the method only requires two quantiles in decile, i.e. the income shares of the bottom 40% and top 10%. The absolute values of the maximum and minimum errors, and

**Table 5. Comparison between the best pairwise and the error minimization methods using PIP and LIS records in the WIID database.**

| Category | Error | PIP records | | | | LIS records | | | |
|---|---|---|---|---|---|---|---|---|---|
| | | (40%,90%) | Model(3) | Model(4) | Model(5) | (40%,90%) | Model(3) | Model(4) | Model(5) |
| All | Max | 0.00679 | 0.00212 | 0.00252 | 0.01514 | 0.00932 | 0.00252 | 0.00246 | 0.01567 |
| | Min | -0.00466 | -0.00806 | -0.00852 | -0.01113 | -0.00591 | -0.00852 | -0.00748 | -0.01565 |
| | RMSE | 0.00180 | 0.00116 | 0.00100 | 0.00701 | 0.00173 | 0.00100 | 0.00093 | 0.00685 |
| Low | Max | 0.00525 | 0.00123 | 0.00119 | 0.01334 | 0.00816 | 0.00119 | 0.00096 | 0.01567 |
| | Min | -0.00328 | -0.00107 | -0.00211 | 0.00629 | -0.00309 | -0.00211 | -0.00228 | 0.00492 |
| | RMSE | 0.00163 | 0.00044 | 0.00055 | 0.00939 | 0.00145 | 0.00055 | 0.00056 | 0.00935 |
| Medium | Max | 0.00679 | 0.00212 | 0.00234 | 0.01514 | 0.00808 | 0.00234 | 0.00235 | 0.01441 |
| | Min | -0.00288 | -0.00230 | -0.00250 | 0.00214 | -0.00365 | -0.00250 | -0.00251 | 0.00123 |
| | RMSE | 0.00191 | 0.00073 | 0.00066 | 0.00736 | 0.00176 | 0.00066 | 0.00067 | 0.00671 |
| High | Max | 0.00513 | 0.00110 | 0.00252 | 0.01224 | 0.00558 | 0.00252 | 0.00246 | 0.00948 |
| | Min | -0.00466 | -0.00446 | -0.00364 | -0.00213 | -0.00591 | -0.00364 | -0.00365 | -0.00168 |
| | RMSE | 0.00183 | 0.00132 | 0.00110 | 0.00443 | 0.00160 | 0.00110 | 0.00111 | 0.00398 |
| Very high | Max | 0.00581 | 0.00165 | 0.00104 | 0.01455 | 0.00932 | 0.00104 | 0.00090 | 0.00609 |
| | Min | -0.00293 | -0.00806 | -0.00852 | -0.01113 | -0.00281 | -0.00852 | -0.00748 | -0.01565 |
| | RMSE | 0.00156 | 0.00252 | 0.00258 | 0.00310 | 0.00240 | 0.00258 | 0.00215 | 0.00420 |

[1]The PIP records come from WIID, data quality level is average. Low, medium, high and very high inequality economies have 506, 1110, 508, 259 records, respectively. There are a total of 2383 records.

[2]The LIS records come from WIID, data quality level is high. Low, medium, high and very high inequality economies have 1139, 3090, 907, 409 records, respectively. There are a total of 5545 records.

the root mean square error between the estimated value and the true value of the Gini coefficient are within a hundredth.

When the income distribution is known, for example, the lognormal distribution or Beta-2 distribution, the best pairwise method is verified valid by using the income samples randomly generated by these distributions. Also, the results of Monte Carlo simulation support the proposition proposed in this paper. When the income distribution is unknown, for verification of the best pairwise method, we use the high-quality quintile data from the World Bank's PIP source and the Luxembourg Income Institute's LIS source in WIID database. We find that the income Gini coefficient depends on the income shares of the bottom 40% and the top 10% of the population, indicating the validity of the pairwise method. Therefore, the best pairwise method provides a practical tool for estimating the sample Gini coefficient.

The best pairwise method bridges the Gini coefficient and the Palma ratio, both of which are determined by the income share of the top 10% and the bottom 40% groups, and contain the same information. This fact suggests that governments and public policy makers must pay attention to the income levels of the bottom 40% and the top 10% groups. In order to narrow the income gap, the income distribution policy should pay attention to raising the income of the bottom 40% group and lowering the income of the top 10% group, for example, setting a minimum wage and the key targets of poverty alleviation. In addition, tax policies can adopt strategies such as progressive tax rates to make high-income groups bear a higher tax burden. In social welfare arrangements, priority should be given to the bottom 40% of the population, such as providing social minimum medical care and old-age security, and special institutional design for the non-income group.

## Author Contributions

**Conceptualization:** Pingsheng Dai.

**Data curation:** Pingsheng Dai.

**Formal analysis:** Sitong Shen.

**Methodology:** Pingsheng Dai, Sitong Shen.

**Software:** Pingsheng Dai.

**Supervision:** Pingsheng Dai.

**Validation:** Sitong Shen.

**Writing – original draft:** Pingsheng Dai.

**Writing – review & editing:** Pingsheng Dai, Sitong Shen.

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
