## [Decision Letter · Decision Letter 0]

13 Dec 2024

PONE-D-24-49350Estimation of the Gini coefficient based on two quantilesPLOS ONE

Dear Dr. Dai,

Thank you for submitting your manuscript to PLOS ONE. After careful consideration, we decided that your article needs a **major** revision. Therefore, we invite you to submit a revised version of the manuscript that addresses the points raised during the review process.

We look forward to receiving your revised manuscript.

Kind regards,

Dr. Mahmut Zeki Akarsu

Academic Editor

PLOS ONE

Reviewers' comments:

Reviewer's Responses to Questions

**Comments to the Author**

1. Is the manuscript technically sound, and do the data support the conclusions?

Reviewer #1: Partly

Reviewer #2: Yes

Reviewer #3: Partly

2. Has the statistical analysis been performed appropriately and rigorously? 

Reviewer #1: Yes

Reviewer #2: Yes

Reviewer #3: No

3. Have the authors made all data underlying the findings in their manuscript fully available?

Reviewer #1: Yes

Reviewer #2: Yes

Reviewer #3: Yes

4. Is the manuscript presented in an intelligible fashion and written in standard English?

Reviewer #1: Yes

Reviewer #2: Yes

Reviewer #3: No

5. Review Comments to the Author

Reviewer #1: The choice of using only two quantiles (top 10% and bottom 40%) to estimate the Gini coefficient is mentioned but not adequately justified. The manuscript lacks a thorough discussion on why this approach, rather than using additional quantiles, is optimal or preferable. Expanding on the rationale or comparison with other methods could strengthen the credibility of this approach.

The modifications to the Lorenz curve (reducing from three parameters to two) are central to the study, yet they’re only briefly described. This simplification needs a deeper explanation, ideally with theoretical or empirical support showing that the two-parameter approach is valid and reliable across different income distributions.

The Monte Carlo simulation used to validate the method is limited in scope. Testing only log-normal and Beta-2 distributions may not capture the diversity of real-world income distributions. Including additional income distribution models could improve the robustness of the validation process.

The manuscript sometimes interchanges terms like "Lorenz fitting curve" and "Lorentz fitting curve" incorrectly, creating potential confusion. Ensuring consistent terminology, particularly around key mathematical constructs, would improve clarity.

While the method is demonstrated with data from specific datasets (PIP and LIS), there’s no discussion on potential limitations or biases these datasets might introduce. Addressing how representative these datasets are (or aren't) for global income inequality would be necessary to evaluate the generalizability of the findings.

The paper focuses heavily on the technical aspects of the estimation without connecting to broader implications. For instance, discussing the practical relevance of this method in policy-making or inequality studies would help contextualize the value of the findings.

The manuscript has an "Ethical Approval" section but lacks a discussion on any data-related limitations, such as privacy concerns or data-sharing issues that may arise. Expanding this section to address data ethics comprehensively would be beneficial, especially given the sensitivity of income data.

The literature review briefly mentions relevant works but lacks a detailed comparison with existing methods for estimating the Gini coefficient, especially those that use more comprehensive data points. This makes it difficult to assess the novelty or improvement this paper’s method offers.

Reviewer #2: 1-Sections and subsection should be numerated according to the journal style.

2- the last paragraph in the introduction should mention all sections in this paper with a brief description about their contents. Also, the word "section" must be "Section" in this paragraph.

3-after Eq(1) the authors must define what is q,p_i. and always express vector in bold in eqs and texts.

4-The references are not new and updated, except few like ref [4,16].

5- In Page 2 Ethical Approval should be mentioned by the end of the paper, so need to be removed.

6- after eq (3) the sentence "y denotes the cumulative..." but there is no y in the above equation.

7-the left hand side of equation 4 contains (a), but no (a) in the right side?

8-For the last paragraph page 5, it is better to write the algorithm as steps to be more obvious for the readers.

9-In the first line on page 12, the process was repeated 100 times, but in statistically to get a robust and more accurate results the replications should be more, like 1000, or 5000. So why the authors used very small replications, are there any obstacles during performing simulation analysis?

10- The name of the software program that was already used for simulation and application part should be clearly mentioned with its version.

11- For the results in table 2, and 3, it is more convenient to use large sample sizes like 500, 1000, and so on. My query is about smaller samples like 100 or 200. does it give same results? did the authors check that?

12- English Editing is recommended

Reviewer #3: Major issues:

The contribution of the paper is not well defined and the background about Lorentz curve and Gini coefficient of inequality is not illustrated efficaciously in the introduction. The literature review on the topic should be expanded, as only a few contributions are considered.

Moreover,

Equation(1): "f is the Dirichlet joint distribution function": which are the random components for which f is the joint df? In the equation, I find f(q|theta)

Equation (3): a, p and q are not defined

Equation (4): the parameter a is not contained in the expression of L

Proposition 1 should be better framed by introducing the necessary definitions for its understanding.

Minor issues:

There are some typos throughout the paper, for instance:

End of the Introduction: "the parameters of the Lorenz fitting curve is estimated in section 2"

The deciles of data are most common (N=10), so consider pairing 10%, 20%, 30%,. . . , 70%, 80%, 90%." is unclear

The ethical approval should be move to the end of the manuscript.

I would repalce "Professor Palma of Harvard University believes that..." with "In [20], the author believes that..."

Review the punctuation.

6. PLOS authors have the option to publish the peer review history of their article (what does this mean?). If published, this will include your full peer review and any attached files.

Reviewer #1: No

Reviewer #2: No

Reviewer #3: No

---

## [Author Response · Author response to Decision Letter 0]

8 Jan 2025

Response to Editor and Reviewers

Editor

Thank you for uploading your study’s underlying data set. Unfortunately, the repository you have noted in your Data Availability statement does not qualify as an acceptable data repository according to PLOS’s standards.

At this time, please upload the minimal data set necessary to replicate your study’s findings to a stable, public repository (such as figshare or Dryad) and provide us with the relevant URLs, DOIs, or accession numbers that may be used to access these data. For a list of recommended repositories and additional information on PLOS standards for data deposition, please see

https://journals.plos.org/plosone/s/recommended-repositories.

Reply: Thank you very much for the suggestion. Since the recommended repositories, such as figshare or Dryad, are charged, we would like to store our data in a free and public repository, i.e. OSF. The link is as follows: https://osf.io/m5vhg/

Reviewer#1

1.The choice of using only two quantiles (top 10% and bottom 40%) to estimate the Gini coefficient is mentioned but not adequately justified. The manuscript lacks a thorough discussion on why this approach, rather than using additional quantiles, is optimal or preferable. Expanding on the rationale or comparison with other methods could strengthen the credibility of this approach.

Reply: Thank you for this pertinent comment. This paper is theoretically based on the work of Palma (2011, 2014) and Cobham and Sumner (2012). They developed the notions of the Palma proposition and Palma ratio. According to these notions, the income inequality can be approximately estimated by using two quantiles (for instance, the top 10% and the bottom 40%). The best pairwise method has undergone proposition refinement in Table 1, and comparisons of different quantile pairings are provided in Tables 2 and 3 (P5, 7, 8). Table 1 in the article presents the results of estimating the deviation between the Gini coefficient and the target value for 16 economies in several paired scenarios (36 pairs were generated from 9 quantiles), and the top 10% and bottom 40% quantiles were identified as the best pairs. Tables 2 and 3 compare the estimation performance of Gini coefficients for different pairings through Monte Carlo simulation, and explore the optimal pairing method in depth. We find that among various pairings of two quantiles, the pairing of the top 10% and bottom 40% is the best.

2. The modifications to the Lorenz curve (reducing from three parameters to two) are central to the study, yet they’re only briefly described. This simplification needs a deeper explanation, ideally with theoretical or empirical support showing that the two-parameter approach is valid and reliable across different income distributions.

Reply: Thank you for the valuable comments. We have deeply discussed the reasons of reducing the three-parameter-Lorenz curve to two-parameter one. Please see the colored context on page 3. We hope the revision can help alleviate the reviewer’s concerns about the validity of our method.

3. The Monte Carlo simulation used to validate the method is limited in scope. Testing only log-normal and Beta-2 distributions may not capture the diversity of real-world income distributions. Including additional income distribution models could improve the robustness of the validation process.

Reply: Thank you for this good advice. The new method builds on the Palma proposition and can be applicable to all income distributions in the real world. In order to overcome the deficiencies in selecting income distributions in Monte Carlo simulations, we use all records from the World Bank PIP and Luxembourg Income Studies LIS, which can reflect the diversity of world income distributions. The results show that the best pairing method is effective and robust (please see Table 5 on page 10).

4. The manuscript sometimes interchanges terms like "Lorenz fitting curve" and "Lorentz fitting curve" incorrectly, creating potential confusion. Ensuring consistent terminology, particularly around key mathematical constructs, would improve clarity.

Reply: Thank you for this nice advice. We have checked the manuscript carefully, and corrected errors in spelling, signs in mathematical formulas, etc.

5. While the method is demonstrated with data from specific datasets (PIP and LIS), there’s no discussion on potential limitations or biases these datasets might introduce. Addressing how representative these datasets are (or aren't) for global income inequality would be necessary to evaluate the generalizability of the findings.

Reply: Thanks a lot for this pertinent advice. The records in the WIID database are sourced from surveys conducted by the World Bank, Luxembourg Institute of Income Research, and national or regional statistical agencies, and the data quality varies. Due to our need for the correct Gini coefficient as the target value, it is required that each record not only has complete decile data, but also has the corresponding correct Gini coefficient. The WIID database provides four categories of data quality levels for each record: high, average, low, and unknown. We extracted all records from the PIP and the LIS, the former has an average the data quality, and the latter a high level of data quality. Since both PIP and LIS record comprehensive data from economies around the world, we hope the used data can minimize the potential limitations or biases.

6. The paper focuses heavily on the technical aspects of the estimation without connecting to broader implications. For instance, discussing the practical relevance of this method in policy-making or inequality studies would help contextualize the value of the findings.

The manuscript has an "Ethical Approval" section but lacks a discussion on any data-related limitations, such as privacy concerns or data-sharing issues that may arise. Expanding this section to address data ethics comprehensively would be beneficial, especially given the sensitivity of income data.

Reply: Thank you very much for this good suggestion. We have added the practical implications of our method for policy-making and academic research in the revised manuscript (please see the colored contexts in the last paragraph on page 11).

The data in this article is entirely sourced from the database of the World Institute for Development Economics Research at the United Nations University (UNU-WIID), which includes surveys conducted by the World Bank, the Luxembourg Institute of Income Research, and national or regional statistical agencies. These data are publicly available, and measured at country (or region)-level, so there is no the ethical issues, such as privacy concerns and sensitivity.

7. The literature review briefly mentions relevant works but lacks a detailed comparison with existing methods for estimating the Gini coefficient, especially those that use more comprehensive data points. This makes it difficult to assess the novelty or improvement this paper’s method offers.

Reply: Thank you for the reviewer's comments and constructive suggestions. We have added a comparison of the error between the best pairing method and the error minimization technique in estimating the Gini coefficient results in Table 5 on Page 10. Based on models (3), (4), and (5), the error minimization technique (using entire data points) is used to estimate the Gini coefficient. The results indicate that the best pairing method is slightly inferior to the 3-parameter and 4-parameter models based on error minimization techniques, but superior to the 2-parameter model.

Reviewer #2:

1.Sections and subsection should be numerated according to the journal style.

Reply: Thank you very much for this suggestion. We have made corresponding modifications based on the journal style.

2. The last paragraph in the introduction should mention all sections in this paper with a brief description about their contents. Also, the word "section" must be "Section" in this paragraph.

Reply: Thank you very much for the advice. We have made corresponding revision according to the advice.

3. After Eq(1) the authors must define what is q,pi. and always express vector in bold in eqs and texts.

Reply: Thank you for the advice. We have defined the meanings of qi and pi in the first paragraph on page 2. We also have made modifications of the expressions of vector. In all equations involved the vectors θ and q, the vectors are written in bold format, and the necessary revisions in the text have also been made.

4. The references are not new and updated, except few like ref [4,16].

Reply: Thank you for the comment. We have added 2 literatures published during the past two years, please see the references [19] [25].

5. In Page 2 Ethical Approval should be mentioned by the end of the paper, so need to be removed.

Thank you for the reviewer’s comments. We have moved the Ethical approval to the end of the paper.

6. After eq (3) the sentence "y denotes the cumulative..." but there is no y in the above equation.

Reply: Thank you for pointing out this error. The y here is the y-axis of the fitted curve, which is corresponding to the L in equation (3). We have changed y to L in the revised manuscript.

7. The left hand side of equation 4 contains (a), but no (a) in the right side?

Reply: Thank you for pointing out this error. The parameter, a, should not appear in equation (4). We have corrected this error.

8. For the last paragraph page 5, it is better to write the algorithm as steps to be more obvious for the readers.

Reply: Thank you very much for the advice. We wrote out the steps of the algorithm in the revised manuscript. Please see the last paragraph on page 6 and the first paragraph on page 7.

9. In the first line on page 12, the process was repeated 100 times, but in statistically to get a robust and more accurate results the replications should be more, like 1000, or 5000. So why the authors used very small replications, are there any obstacles during performing simulation analysis?

Reply: Thank you very much for the advice. According to the reviewer’s suggestion, we increased the number of repetitions to 1000. For each pairing, we generated 1000 sets of decile data. The new results are reported in Tables 2 and 3. We found a little changes in the results, but it does not alter the basic conclusion. Please refer to Tables 3 and 4 on Pages 7 and 8.

10. The name of the software program that was already used for simulation and application part should be clearly mentioned with its version.

Reply: Thank you for the suggestions. We used EVIEWS10.0 software for data programming and processing, and we have pointed out this on Page 6 in the manuscript.

11. For the results in table 2, and 3, it is more convenient to use large sample sizes like 500, 1000, and so on. My query is about smaller samples like 100 or 200. does it give same results? did the authors check that?

Reply: Thank you for the comments. This is involved the problem about sample boundaries. We also simulated smaller sample sizes such as 100 and 200. As mentioned in the article, the estimating accuracy of the Gini coefficient increases with the increase of sample size. The following table reports the result of the simulation with 100 sample size. We find that the accuracy is not good as that of simulation with 500 or bigger sample size. Specifically, when the sample sizes are equal to 100 or 200, the estimation accuracy of the Gini coefficient decreases, and some cases that do not meet the requirements of effective estimation may occur. We have added an explanation for small sample size in the first paragraph on Page 8. The specific sample boundaries will need further exploration in the future.

Obj. Income distribution Error Pairwises estimation error (1000 repetitions)

 (90%,40%) (80%,30%) (90%,50%) (80%,40%) (70%,30%) (60%,40%)

100 Log-normal

Max 0.01614 0.00686 0.01365 0.00456 0.00971 0.02334 

 Min -0.01627 -0.02322 -0.01558 -0.02231 -0.03730 -0.04882 

 RMSE 0.00627 0.00855 0.00664 0.00881 0.01336 0.01795 

 Beta-2 

Max 0.01229 0.01279 0.01822 0.01875 0.02936 0.04085 

 Min -0.00907 -0.01197 -0.00284 -0.00204 -0.00706 -0.01255 

 RMSE 0.00479 0.00516 0.01141 0.00764 0.01111 0.02265 

12. English Editing is recommended.

Reply: Thank you for the advice. We have invited a professional English editor to proofread this manuscript in terms of English grammar, spelling, etc. Some minor issues have been corrected.

Reviewer #3: Major issues:

1.The contribution of the paper is not well defined and the background about Lorentz curve and Gini coefficient of inequality is not illustrated efficaciously in the introduction. The literature review on the topic should be expanded, as only a few contributions are considered.

Reply: Thank you for the reviewer’s comments and constructive suggestions. Firstly, in the last paragraph on page 2, we have refined the contributions of this paper. Secondly, in the introduction section, we introduced the foundation for the background of the Lorenz curve and Gini coefficient. Please refer to the first paragraph on Page 1. Finally, we have expanded the literature review, please see the third paragraph on page 2.

2. Moreover, Equation(1): "f is the Dirichlet joint distribution function": which are the random components for which f is the joint df ? In the equation, I find f(q|theta).

Reply: Thank you for the comment. The Dirichlet function f is the joint distribution of the income shares qi of N equal intervals, pi is the population share accumulated to the i-th equal point (i=1,2,..., N), q is the vector (q1, q2,..., qN) T, and θ is the parameter vector of the Lorentz curve.

3. Equation (3): a, p and q are not defined.

Reply: Thank you for the comment. Due to our negligence, we mistakenly wrote L as y in Equation (3) in the original submission. We have corrected this error. We have also added the definitions for the parameters a, p, and q. Specifically, a is the scale parameter of the Lorenz curve, p and q are the shape parameters of the Lorenz curve.

4. Equation (4): the parameter a is not contained in the expression of L. Proposition 1 should be better framed by introducing the necessary definitions for its understanding.

Reply:Thank you for pointing out the error. Equation (4) should not include the parameter a. We have corrected this error. In the revised manuscript, a definition and a proposition are added for helping understand Equation (4). Please see the colored paragraphs on pages 5 and 6.

Minor issues:

1. There are some typos throughout the paper, for instance: End of the Introduction: "the parameters of the Lorenz fitting curve is estimated in section 2".

Reply: Thank you for pointing out the typos. We have corrected the error that the reviewer has pointed out here. We have also proofread the full text and revised similar errors.

2. The deciles of data are most common (N=10), so consider pairing 10%, 20%, 30%,. . . , 70%, 80%, 90%." is unclear.

Reply: Thank you for this good comment. We have rewritten this sentence as “The decile data are most common, so consider available pairings among 10%, 20%, 30%, ..., 70%, 80%, and 90% quantiles, there are 36 combinations in total.” 

3. The ethical approval should be move to the end of the manuscript.

Reply: Thank you for this advice. We have moved 'The ethical approval' to the end of the manuscript.

4. I would repalce "Professor Palma of Harvard University believes that..." with "In [20], the author believes that..."

Reply: Thanks a lot. We have rewritten this sentence on Page 2 according to this advice.

5. Review the punctuation.

Reply: Thank you for this advice. We have carefully reviewed the punctuations, and modified some errors.

---

## [Decision Letter · Decision Letter 1]

23 Jan 2025

Estimation of the Gini coefficient based on two quantiles

PONE-D-24-49350R1

Dear Dr. Pingsheng Dai,

We’re pleased to inform you that your manuscript has been judged scientifically suitable for publication and will be formally accepted for publication once it meets all outstanding technical requirements.

Kind regards,

Dr. Mahmut Zeki Akarsu

Academic Editor

PLOS ONE

Reviewers' comments:

**Comments to the Author**

1. If the authors have adequately addressed your comments raised in a previous round of review and you feel that this manuscript is now acceptable for publication, you may indicate that here to bypass the “Comments to the Author” section, enter your conflict of interest statement in the “Confidential to Editor” section, and submit your "Accept" recommendation.

Reviewer #1: All comments have been addressed

Reviewer #2: All comments have been addressed

2. Is the manuscript technically sound, and do the data support the conclusions?

Reviewer #1: Yes

Reviewer #2: Yes

3. Has the statistical analysis been performed appropriately and rigorously? 

Reviewer #1: Yes

Reviewer #2: Yes

4. Have the authors made all data underlying the findings in their manuscript fully available?

Reviewer #1: Yes

Reviewer #2: Yes

5. Is the manuscript presented in an intelligible fashion and written in standard English?

Reviewer #1: Yes

Reviewer #2: Yes

6. Review Comments to the Author

Reviewer #1: Excellent revision by the author; I accept the article in its current form. I hope this version is excellent.

Reviewer #2: After a second round revision, i found that the authors addressed all my comments and modified the manuscript, hence, it can be published in its current format.

7. PLOS authors have the option to publish the peer review history of their article (what does this mean?). If published, this will include your full peer review and any attached files.

Reviewer #1: **Yes: **Dr Muhammad Ameeq

Reviewer #2: No

---

## [Editor Report · Acceptance letter]

30 Jan 2025

PONE-D-24-49350R1 

PLOS ONE

Dear Dr. Dai, 

I'm pleased to inform you that your manuscript has been deemed suitable for publication in PLOS ONE. Congratulations! Your manuscript is now being handed over to our production team.

Kind regards, 

on behalf of

Dr. Mahmut Zeki Akarsu 

Academic Editor

PLOS ONE